# Experimental noise cutoff boosts inferability of transcriptional networks in large-scale gene-deletion studies

C.F. Blum[1,2], N. Heramvand[1,2], A.S. Khonsari[1] & M. Kollmann[1]

Generating a comprehensive map of molecular interactions in living cells is difficult and great efforts are undertaken to infer molecular interactions from large-scale perturbation experiments. Here, we develop the analytical and numerical tools to quantify the fundamental limits for inferring transcriptional networks from gene knockout screens and introduce a network inference method that is unbiased with respect to measurement noise and scalable to large network sizes. We show that network asymmetry, knockout coverage and measurement noise are central determinants that limit prediction accuracy, whereas the knowledge about gene-specific variability among biological replicates can be used to eliminate noise-sensitive nodes and thereby boost the performance of network inference algorithms.

[1] Institute for Mathematical Modeling of Biological Systems, Heinrich-Heine University of Düsseldorf, Universitätsstraße 1, 40225 Düsseldorf, Germany. [2] Max Planck Institute for Plant Breeding Research, Carl-von-Linné-Weg 10, 50829 Köln, Germany. C.F. Blum and N. Heramvand contributed equally to this work. Correspondence and requests for materials should be addressed to M.K. (email: markus.kollmann@hhu.de)

The functionality of a living cell is determined by the interplay of multiple molecular components that interact with each other. Generating a global map of these molecular interactions is an essential step to advance our understanding of the molecular mechanisms behind disease, development and the reprogramming of organisms for biotechnological applications[1]. The current advances in gene-editing methods[2] have scaled up the size of genome-wide single and double knockout libraries, ranging from microbes[3, 4] to higher eukaryotes[5] and open up a much more informative data source than inferring gene-regulatory networks from unspecific perturbations, such as stress or changes in growth conditions[6]. However, the detection of direct interactions between two genes from association measures–for example, the covariance between transcript levels–remains a highly non-trivial task, given the significant variation among biological replicates, the frequent case where the number of parameters exceeds the number of independent data points, and the high dimensionality of the inference problem. In addition, direct interactions inferred from transcriptome data typically oversimplify the molecular complexity behind gene regulation, which frequently involves protein–protein interactions and modifications on protein or DNA level. Consequently, gene interaction networks inferred from transcriptome studies should in general not be interpreted as or compared with gene-regulatory networks. In this work we first investigate the causes that affect network inferability by introducing a simple network inferability measure and subsequently use the gained insight to design an unbiased, scalable network inference algorithm.

## Results

**Network inferability**. The existence of a direct interaction between gene A as source of regulation (source node) and gene B as target of regulation (target node) can be detected if a significant part of the transcriptional activity of B can be explained by the transcriptional activity of A but not by the transcriptonal activities of the remaining genes in the network. Thus, a necessary condition for identifiability or inferability of links is the knowledge about the information that can be transmitted by alternative routes in the network, which can be obtained by targeted, external perturbations of node activities[7]. As most gene perturbation screens are incomplete—for example, owing to the fact that essential genes cannot be knocked out—we have in general the situation that a significant amount of interactions within an $N$-gene network are non-inferable, regardless of the amount of experimental replicates and the strength of perturbations.

**Limits of network inferability**. To estimate the upper bound of links that can be inferred from knockout screens, we consider a directed but not necessarily acyclic network of $N$ nodes, with node activities as observables and a predefined subset of nodes that are perturbed independently by external forces. The perturbed nodes are randomly distributed within the network and we denote by $q$ the fraction of nodes that are perturbed. We assume that an arbitrarily large set of perturbation experiments can be generated, with the freedom to tune the perturbation strength for each node independently. We further assume that other perturbative sources and measurement noise are absent. Calculation of the expected fraction of inferable links, $F(q)$, can be carried out by a simple counting procedure (Figs 1a and 2a), assuming that links can be represented by noiseless, linear functions with non-zero slope. Under these conditions, a directed link between source and target node is inferable–or equivalently its link strength is identifiable–if it is not possible to fully reconstruct the activity state of the source node from the node activities of the remaining network. Consequently, a link is inferable if a part of the variation of the target node can be only explained by the source node, given that a link between them exists, and implies non-zero partial correlation between source and target node. To allow detection of arbitrarily small partial correlations, we make sure that there exists a finite fraction of experiments for each target node, where the target node is not perturbed (Online Methods and Supplementary Note 1). If, for example, only one node in the network is perturbed that targets multiple other nodes, its node activity can be fully reconstructed by any of its targets, resulting in zero partial correlation coefficients, which implies that none of the directed links can be inferred (Fig. 1a, right network). In contrast, if two out of three nodes are perturbed, all links targeting the unperturbed node are inferable (Fig. 1a, left network). In addition, nodes that have been identified as targets of the current target node can be removed prior to inference. This is because an existing link from the actual target node excludes them from

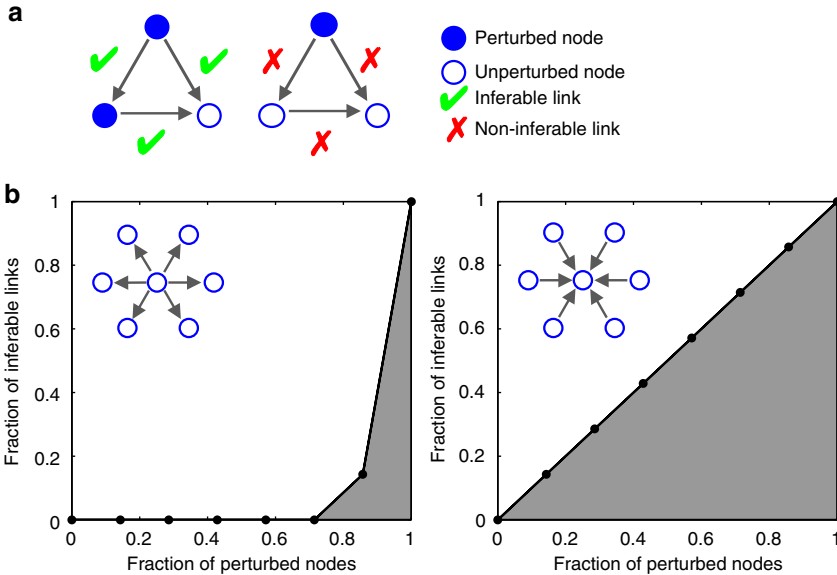

**Fig. 1** Illustrative example of network inferability. **a** Left network: fully inferable network; Right network: non-inferable network. **b** Fraction of inferable links versus fraction of perturbed nodes in the network. Left panel: hub of outgoing nodes. Right panel: hub of incoming nodes

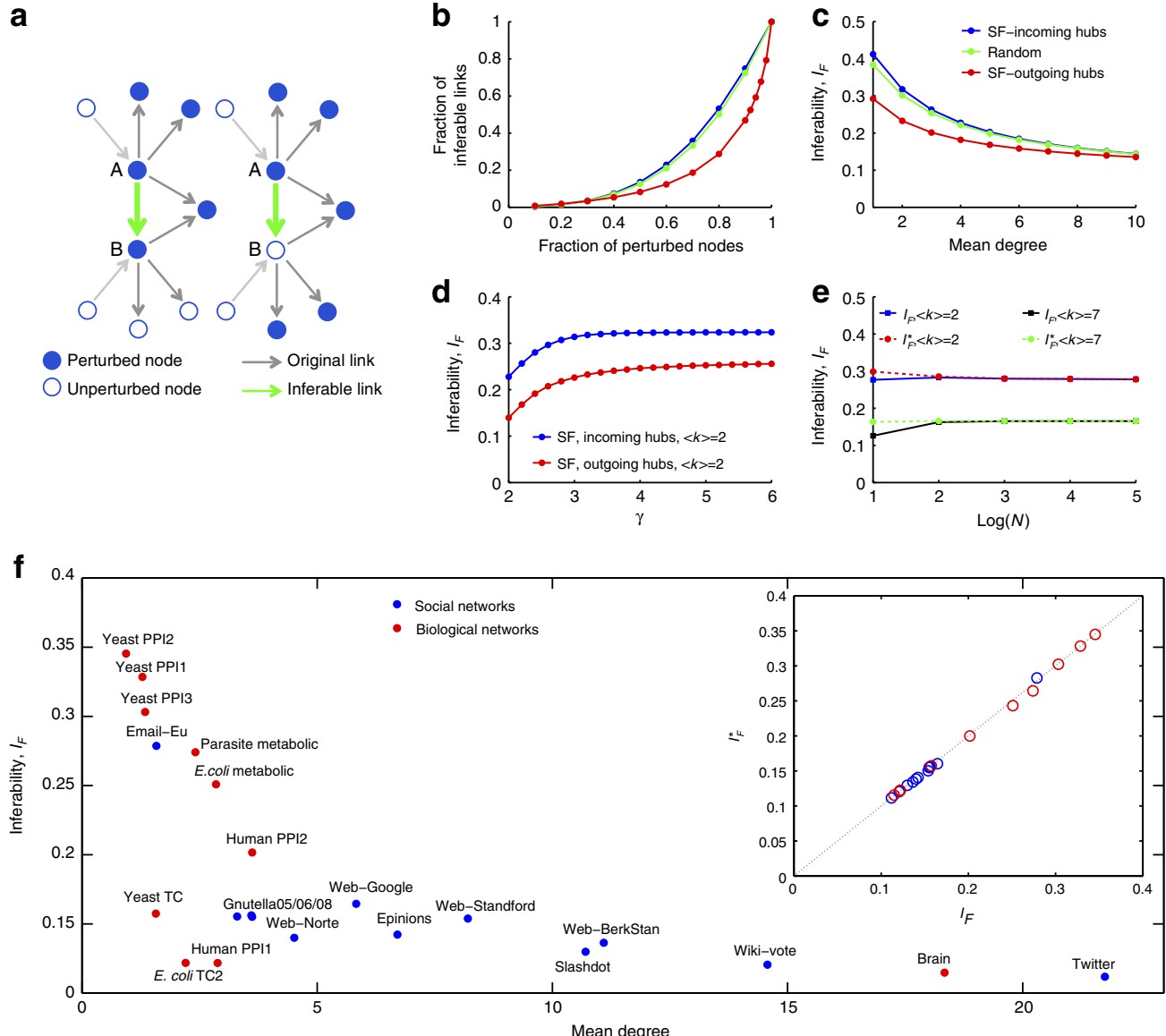

**Fig. 2** Inferability as a function of network parameters. **a** Directed links are inferable if either all outgoing links of the source node point to perturbed nodes including the target node (left panel) or if all outgoing links of source node and target node point to perturbed nodes, with the target node not perturbed (right panel). **b** Fraction of inferable links against the fraction of perturbed nodes using three network types: (i) scale-free network with exponent $\gamma = 2.5$ and mean degree $\langle k \rangle = 3$, where nodes of higher degree target nodes of lower degree (outgoing hubs), (ii) same network as (i) but with all link directions inverted (incoming hubs) or a network generated by random insertion of links with the same mean degree as scale-free networks (random network). Colour coding as in **c**. **c** Network inferability versus mean degree, using networks of **b**. **d** Network inferability versus scaling exponent for two types of scale-free networks. **e** Asymptotic invariance of the two inferability measures introduced in the main text with respect to network size. **f** Network inferability as a function of mean degree for social and biological networks. Correlation between the two inferability measures introduced in the main text (inset)

transmitting information back to it, as we exclude bidirectional links from our analysis. This makes the network (Fig. 1a, left network) fully inferable, as the link between the remaining two perturbed nodes can be inferred by collecting experiments for which the target node is unperturbed. We emphasise that our approach to network inferability does not account for a priori known restrictions on the network topology, as in the case of directed acyclic graphs. Such constraints can strongly increase the inferability of directed links[8].

As $F(q)$ is an upper bound for the expected number of directed links that can be inferred from stationary node activities in the absence of noise and other constraints on the network structure, we now ask how this bound is related to the structural properties of the network. To compare different network architectures, it is

useful to define the network inferability, $I_F$, as the area under the $F(q)$-curve, $I_F := \int_0^1 F(q)\,\mathrm{d}q$. Comparison of $I_F$ between two general classes of network structures with node degrees either power law or Poisson distributed shows that networks that are enriched with nodes of high outdegree are the most difficult ones to infer (Figs 1b and 2b). The reason is that whenever hubs with high outdegree are perturbed there is a high chance that they target more than one of the unperturbed nodes and without additional perturbations it is impossible to detect which of the targets are affected directly and which indirectly. Differences in inferability due to network structure are most predominant for networks with low mean degree and become less predominant with high mean degree (Fig. 2c). As our measure of inferablity, $I_F$, is essentially determined by the outdegree distribution, the curve

starts saturating for scale-free exponents $\gamma > 3$, as in this regime the variance of the number of links per node is essentially constant for increasing $\gamma$ and fixed mean degree[9] (Fig. 2d). The network inferability, $I_F$, is asymptotically independent of network size (Fig. 2e). We further investigated the inferability of causal interactions in biological and social networks as a function of the mean degree (Fig. 2f). The decreasing trend can be explained by the higher number of alternative routes that come with a more strongly connected network. The low inferability of gene-regulatory networks can be attributed to master regulators that regulate a large fraction of the genome (hubs with high outdegree), whereas the comparatively high inferability of protein interaction networks is a consequence of the low number of different binding domains per protein and that only a fraction of the existing interactions have been identified due to limitations of experimental methods[10]. If we assume that the conditional probability $P(k, l, m|k \to l)$ of finding two connected nodes in the directed network, where the source node has $k \geq 1$ outgoing links, the target node has $l \geq 0$ outgoing links, and both share $m$ nodes as common targets of their outgoing links, can be factorised, the resulting inferability measure, $I_F^*$, is simply a function of the outdegree distributions, $P(k)$ and $P(l)$. We observed that $I_F^* \approx I_F$ for all networks investigated in this work (Fig. 2f, inset). This result shows that for a large variety of networks structures the outdegree is the dominating factor that determines network inferability. Consequently, if the perturbed nodes are not selected at random but are biased towards higher outdegree, inferability is further reduced.

**Network inference concepts**. From our analyses of network inferability we gained the insight that the number of potential alternative routes how a source node can affect a target node correlates positively with the outdegree of the source node and inversely with the expected inferability of the directed link between source and target, given that perturbed nodes are uniformly distributed in the network. Consequently, network inference algorithms should strongly benefit from an a priori reduction in the number of alternative routes. In the following we present an unbiased network inference algorithm that eliminates alternative routes with low signal-to-noise ratio as a preprocessing step. Inference of transcriptional networks on genome scale is best realised by methods that are (i) asymptotically unbiased, (ii) scalable to large network sizes, (iii) sensitive to feed-forward loops[11] and (iv) can handle data sets with and without knowledge about which nodes are targeted by experimentally induced perturbations[7, 12–16] (Supplementary Note 2). Inference methods for directed networks typically require individual perturbation of all nodes[7] or many perturbations of different strengths to compute conditional association measures[6, 17] or conditional probabilities[18]. Generation of time course data seems to be the most natural way to infer directed networks by simply analysing the temporal ordering of the transcriptional activities[19, 20]. However, this approach precludes the use of knockout experiments and requires fast acting perturbations in combination with monitoring node activities over time, which is experimentally demanding, especially if nodes respond on very different time scales[21].

**Experimental variability and technical noise**. Inference is further complicated by the fact that transcriptome data contain a significant amount of stochastic variation between biological replicates despite pooling over millions of cells (Fig. 3a). It is interesting to see that the variation across biological replicates for baker's yeast[3] is close to a normal distribution and follows almost exactly a $t$-distribution with 11 degrees of freedom over five standard deviations (Fig. 3a, inset). The same data set also shows

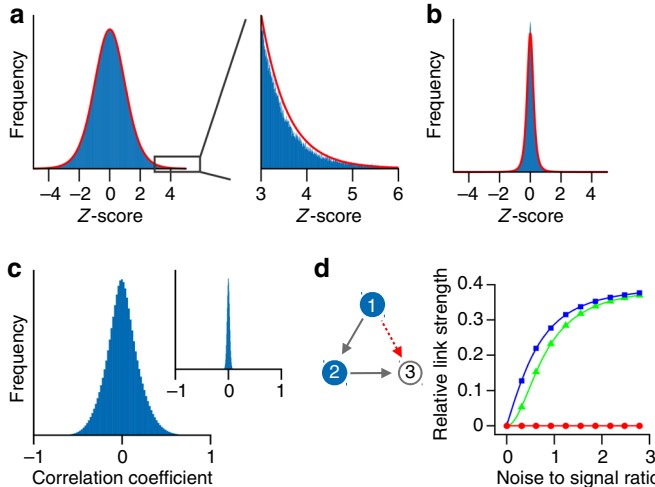

**Fig. 3** Distribution of wild-type expression levels for *S. cerevisiae* from 748 biological replicates. **a** Distribution of the relative expression, $\log_2(r_i)$, with $r_i := x_i/x_i^{pool}$ and $x_i$ the expression of gene $i$ relative to $x_i^{pool}$ followed by standardisation of the $\log_2$ fold changes ($z$-score). The values $x_i^{pool}$ have been obtained by first pooling the 748 biological replicates before measuring gene expression. The distribution is well described within five standard deviations by a $t$-distribution with 11 degrees of freedom (red line). **b** Distribution as in **a** but now for differences among technical replicates. **c** Correlation between gene expression levels is significantly higher than expected by chance (inset) **d** Illustration of a noise induced false positive link (red arrow) as described in the main text. Data for a three-node network with two links was generated by applying independent perturbations on node 1 and node 2. The link strength of the non-existing link from node 1 to node 3 relative to the existing link from node 2 to node 3 was inferred using three different methods (i) partial correlations (blue squares), (ii) conditional mutual information (green triangles) and PRC (red circles)

that variability among biological replicates is much larger than technical noise (Fig. 3b) for this experimental setup. As variability among biological replicates may arise from subtle differences in growth conditions that induce changes in gene regulation, we expected to see significant cross-correlations among genes (Fig. 3c), whose magnitude is much larger than expected by chance (Fig. 3c, inset). These cross-correlations can be exploited for inferring the structure of undirected networks[12], if the driving noise is independent and identically distributed for all nodes (Supplementary Note 2). In contrast, technical noise not only reduces the statistical significance for detecting true interactions but can also induce a significant fraction of false positive interactions, especially if the interaction network under investigation is sparse. Such noise induced misclassification of links can be illustrated by a simple linear network $A \to B \to C$ for which standard inference methods—such as partial correlations–interpret the information that $A$ has about $C$ erroneously as a direct link between $A$ and $C$ if the state of $B$ is corrupted by measurement noise (Fig. 3d). The reason is that a part of the correlation between $A$ and $C$ cannot be explained by $B$.

**Algorithm for large-scale and unbiased network inference**. To make use of the rapidly growing amount of single-gene knockout screens for which transcriptome data are[3] or may become available[22, 23], we developed a method to infer directed networks on a genome scale, where the number of genetic perturbations is typically below the number of nodes or genes in the network (Online Methods). In brief, our method uses the concept of

probabilistic principle component analysis[24] to compute partial response coefficients (PRC) that are asymptotically unbiased with respect to Gaussian measurement noise. In addition, the algorithm provides a feature to identify non-inferable links, which are removed before statistical analysis. In the absence of noise, our numerical method correctly predicts the fraction of links that are inferable, $F(q)$ (Supplementary Note 1), for a network with links represented by linear functions of slope one. To evaluate the performance of our method we generated two synthetic knockout data sets that closely resemble the gene-regulatory network structure of baker's yeast, using the GeneNetWeaver software[25] that uses a hierarchical network structure and our own generative model that uses a scale-free network structure (Supplementary Note 3). We added Gaussian measurement noise to the synthetic data with a standard deviation of 10% the $\log_2$ fold-change in expression level for each perturbation for each gene. Residual bootstrapping among replicates was used to quantify the statistical significance of the inferred link strengths. In comparison with standard inference methods, such as partial correlations[12–14, 26, 27], our method shows a significantly higher performance in the absence of any penalties that enforce sparse network structures (Fig. 4b, left panel). The improved performance of our

approach can be assigned to the fact that the method is unbiased with respect to measurement noise (Online Methods).

To further improve the predictive power of our method we included the prior knowledge that transcriptional networks are highly sparse. Sparsity constraints are typically realised by penalising either the existence of links or the link strengths by adding appropriate cost functions, such as $L^1$-norm regularised regression (Lasso)[28]. Adding a cost function to the main objective comes with the problem to trade-off the log-likelihood against the number of links in the network whose strength is allowed to be non-zero. In the absence of experimentally verified interactions there is no obvious way how to determine a suitable regularisation parameter that weights the likelihood against the cost function, which is one of the great weaknesses of such methods.

In our approach we reduce network complexity by assuming that functionally relevant information in molecular networks can only pass through nodes whose response to perturbations is significantly above the base line that is given by the variability among biological replicates. The individual noise levels can be estimated from natural variations among wild-type experimental replicates (Fig. 3a). The significance level that removes nodes from the network with low signal-to-noise ratio can be set to a desired false discovery rate. It can be shown that removal of noisy nodes imposes a sparsity constraint on the inference problem (Online Methods). The different steps required to arrive at a list of significant links are illustrated in Fig. 4a. In the first step, genes are grouped in clusters that are co-expressed under all perturbations. These clusters are treated as single network nodes in the subsequent steps. In the second step, only those samples are extracted from the data set that correspond to a perturbation of a chosen gene—the source node—with no other genes perturbed (node 5 in Fig. 4a). From this reduced data set, we identify all nodes in the network that change expression above a given significance level upon perturbing the source node. These significantly responding nodes define a subnetwork for each source node, which is typically much smaller in size than the complete network. In the third step, we collect all perturbation

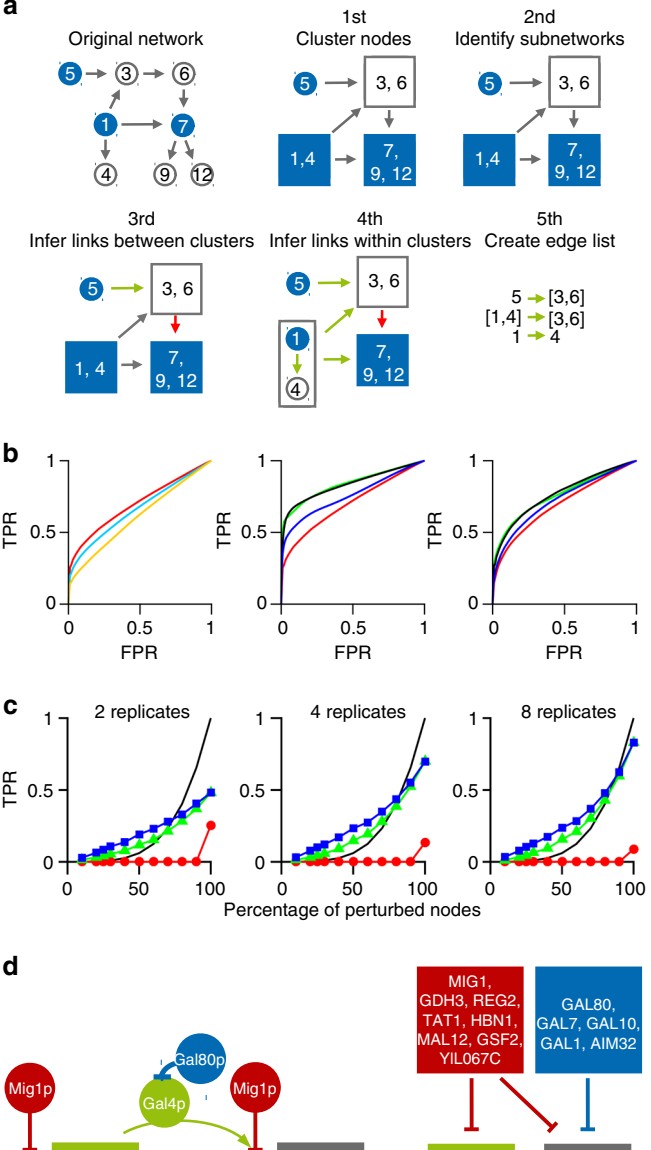

Fig. 4 Performance of our method. **a** Flow-chart showing the algorithmic steps for network inference as explained in the main text. **b** Receiver Operating Characteristic (ROC) curves for 300-node scale-free networks with additive Gaussian measurement noise of 10% of the expression level and 25% of the nodes perturbed. Data were generated using GeneNetWeaver (left and middle panel) as well as using scale-free network structure with mean degree of $\langle k \rangle = 2$ and scaling exponent $\gamma = 2.5$ (right panel, Supplementary Note 3). Here, the true positive rate is computed with respect to the inferable links[39]. Performance of inference methods without sparsity constraints (left panel): PRC (red), partial correlations/linear regression (turquoise) and conditional mutual information (orange). Performance of inference methods with sparsity constraints (middle and right panel): PRC with subnetwork method (green) and Lasso (black) both applied to a subset of significantly responding nodes that were selected with 1% false discovery rate, Lasso regression applied to all 300 nodes (blue), and PRC from left panel (red) for comparison. **c** True positives for the same scale-free network of **b**, with 2, 4 and 8 experimental replicates with 5% false discovery rate for both significantly responding nodes and link strength: PRC (red), PRC with subnetwork method (green), PRC with subnetwork and clustering method (blue), and $F(q)$ (black line). **d** The S. cerevisiae GAL network as an example for a gene-regulatory network where phosphorylated Mig1 sets the basal expression levels of Gal4 and one of its many regulatory targets, Gal3. Gal4 protein can activate Gal3 expression but is inactivated upon binding of Gal80 protein. The transcriptome data set contains knockout mutants for GAL80 and MIG1 but not for the remaining GAL genes. A schematic representation of the key molecular mechanisms (left) and links inferred from transcriptome data[3] (right)

data from the complete data set for all nodes that are part of the subnetwork. Before inferring a direct interaction that points from the source node to a given target node in the subnetwork (green arrows in Fig. 4a), we remove all experiments from the data set where the target node is perturbed. The second and third steps essentially realise the counting procedure of inferable links as illustrated in Fig. 2a, with the difference that significant links are identified by PRCs in combination with residual bootstrapping over replicates (Online Methods, Supplementary Note 3). In the fourth step, we collect all clusters of co-expressed genes that contain exactly two nodes, with one of the nodes perturbed and check statistical significance of the directed link between them. In the fifth step, all significant links are collected in an edge list. We refer to these five steps as the clustering method. If we remove all links from the edge list that have more than one node in a source cluster or more than one node in a target cluster, we obtain an edge list that corresponds to links between single genes. This reduced edge list would also arise by skipping the clustering step and we refer to the remaining inference steps that compute links between single genes as subnetwork method.

**Performance of the proposed inference algorithm**. The Lasso method in combination with bootstrapping has been benchmarked as one of the highest performing network inference methods for in silico generated expression data[29]. The receiver operating characteristic (ROC) curve of the subnetwork method shows better performance than the Lasso method (Fig. 4b, middle and right panel) after adjusting the regularisation parameter of the Lasso method such that the area under the ROC curve is maximised. However, a significant performance boost for the Lasso method can be achieved by applying the second step of our method that removes noisy nodes, resulting in comparable performance of Lasso with the subnetwork method for the case that validation data exist such that the regularisation parameter can be determined (Fig. 4b, middle and right panel).

To get insight into the optimal experimental design for generating data for network inference, we computed the fraction of correctly inferred links and compared them against the fraction of independently perturbed nodes for different numbers of experimental replicates. We compared three different variants of our approach: PRC, PRC together with subnetwork method and PRC together with clustering method (Fig. 4c). As all variants share PRC as underlying inference method (Online Methods), the observed strong increase in performance can be assigned to the sparsity constraint that comes with the subnetwork method or the clustering method. Owing to this constraint, both the subnetwork method and the clustering method can have higher accuracy than the noise-free analytical solution, as the latter does not enforce sparse network structures. The results show that in the presence of 10% measurement noise the amount of available replicates limits the true positive rate, even if 100% of nodes are perturbed. Inference of >80% of the network can only be achieved if the number of replicates is sufficiently high.

To benchmark the performance of our algorithms in comparison to others, we applied our method to the DREAM3

in silico network inference challenge[30]. The provided data set of this challenge has the advantage that full information about the identity of perturbed nodes is given. We ignored the transient information from time series and used the stationary state of time course data to estimate the variation in expression between biological replicates. To identify the significantly responding nodes, we used a Bonferroni corrected significance level of $\alpha = 0.05/N$, where the number of alternative hypotheses—or the number of possible incoming links for a given target node in our case–are bounded by number of possible source nodes in the network, $N - 1$. To make sure that we correctly implemented the published performance evaluation method that is based on curve fitting a sampled null hypothesis[31], we followed the proposed curve fitting procedure suggested by the organisers of the challenge by using different exponential family distributions for each tail, and alternatively by using a single $t$-distribution to fit AUROC null hypothesis samples. The results are shown in Table 1 and Supplementary Data 4. The overall second place among the other 29 inference methods should be interpreted in the light that the better performing algorithm uses extensive hyperparameter tuning, makes use of transient data, and does not scale well with network size[32]. Furthermore, our approach seems to be robust with respect to the chosen significance level as changing $\alpha$ by one order of magnitude did not affect the ranking. However, we emphasise that for 'large $p$ small $n$' problems, where the number of parameters exceeds the number of independent data points, preprocessing often has a larger effect on performance than the inference method itself[30]. For our algorithm the performance boost is a consequence of generating subnetworks as preprocessing step.

**Application to yeast genome knockout data**. To evaluate the performance of our approach on real data, we use one of the largest publicly available transcriptome data sets[3], comprising of transcriptomes that cover 6170 genes for 1441 single-gene knockouts that can be utilised for network inference using PRC. We use the galactose utilisation network as a gene-regulatory example, which is one of the best characterised gene-regulatory modules in yeast[33]. The regulatory mechanism of the GAL4 gene as a key regulator is shown in Fig. 4d, left panel. As information about phosphorylation and protein interaction is absent in expression data, the inferred network structure from transcriptome data with GAL4 and GAL80 perturbed is different from the known gene regulation but can identify major regulators and their targets. Whether the gene AIM32—which is not known to be part of the GAL network—is co-regulated with GAL80 or an artefact of the knockout screen is difficult to judge. Both options are possible as AIM32 is located in close vicinity to GAL80 on the genome. By sorting genes with respect to their number of statistically significant outgoing links, we can identify potential key regulators. Besides transcription factors, the regulators with highest statistical significance are factors involved in chromatin remodelling, signalling kinases, and genes involved in ubiquitination (Supplementary Data 1–3). This result—although expected for eukaryotes—is inaccessible for inference methods

**Table 1 Ranking and overall scores (in paranthese) among the original participants of the DREAM3 in silico network inference challenge**

|  | 10 nodes $\alpha = 0.05/10$ | 50 nodes $\alpha = 0.05/50$ | 100 nodes $\alpha = 0.05/100$ |
| --- | --- | --- | --- |
| Original scoring method | 2nd (4.64) | 2nd (31.43) | 2nd/1st (55.98) |
| AUROC background fitted with $t$-distribution | 2nd (4.14) | 3rd (30.10) | 2nd/1st (50.06) |

Scores were obtained with the original scoring method and a scoring method in which the AUROC background distribution was fitted with a $t$-distribution. Here, $\alpha$ denotes the significance level for the identification of nodes that are significantly affected by perturbations

that a priori fix known transcription factors as regulatory sources. However, as the number of deleted genes in this data set comprise just 23% of the genes for which transcript levels have been measured, we can estimate from our simulations that we have inferred <10% of the direct interactions in the transcriptional network of yeast.

## Discussion

We have developed an unbiased network inference method for perturbation experiments that target individual nodes in the network. Consequently, node activity data that result from unspecific perturbations cannot be exploited by this algorithm in its present form. As individual gene knockout or knockdowns dominate many large-scale experimental studies of node activities in biological networks[3, 23] and their genome-wide coverage is constantly improving[22, 34], we expect that the biological data sets to which the algorithm can be applied will rapidly increase in the near future. However, currently most of the large-scale knockout or knockdown screens lack complete coverage of mutants and often come with low number of experimental replicates, if any. In this work we have shown that insufficient coverage of perturbed nodes in transcriptome data fundamentally limit the amount of links that can be inferred, independently of the employed inference algorithm and that high statistical power requires a significant amount of replicates to drive down effects of experimental variability and measurement noise. We therefore introduced a network inference approach that is able to detect significant links for the case that only a fraction of nodes are perturbed, removes nodes with low signal-to-noise ratio from the network, and makes use of an inference algorithm that is insensitive to measurement noise. Including prior knowledge about network complexity and reducing the effects of noise is crucial for network inference problems, where the number of parameters, e.g., link strengths, scale quadratically with network size and often exceed the number of measured data points. Good scaling behaviour and the absence of time-consuming hyperparameter tuning make our approach an easily applicable network inference tool that shows competitive performance with state-of-the-art methods. However, even when complete coverage of single-gene perturbations together with a high number of experimental replicates of transcriptome data are available, the inferred transcriptional network cannot be directly translated into a model that reflects the biochemical reality of gene regulation. The reason is that gene regulation can involve complex molecular interactions on DNA, RNA, protein and small molecule level that result in direct interactions between mRNA levels. An example of such complex interactions is the observed regulation by the human oncogene IDH1—a metabolic enzyme involved in the citric acid cycle. Mutational loss of normal enzymatic function of IDH1 and production of the metabolite 2-hydroxyglutarate can affect the activity of an epigenetic regulator, which promotes tumorgenesis by reprogramming transcriptional activity on genome scale[35]. Inference of such complex molecular interactions would require a combination of different high-throughput technologies, with the challenge that different methods typically show large differences in sensitivity and coverage[36].

## Methods

**PRC.** We aim at inferring direct interactions between $N$ observable molecular components, such as transcripts or proteins, by measuring their concentrations. We define by $y \in \mathbb{R}^N$ an $N$ dimensional vector that represents the logarithm of these concentrations, which is the natural scale where experimental data are reported. We assume that the available data set has been generated from $P$ perturbation experiments, $\{y_k\}_{k=1}^P$, which may also include experimental replicates. We further assume that the molecular targets of the perturbations are known, as it is the case for gene knockout or knockdown experiments. The elements of the interaction matrix $A \in \mathbb{R}^{N \times N}$ define the strengths of the directed interactions

among the molecular components, for example, $A_{ij}$ quantifies the direct impact of component $j$ on component $i$. Given the available experimental data, our aim is to correctly classify the off-diagonal elements of $A$ as zero or non-zero to obtain the structural organisation of the interaction network. We assume that the observed component abundance on log-scale, $y^{obs}$, differs from the true value, $y$, by additive measurement noise, $\epsilon$, which is characterised by zero mean, $\mathbb{E}[\epsilon] = 0$, and variance, $\mathbb{E}[\epsilon\epsilon^T] = \sigma^2 I_N$, with $I_N$ the $N$ dimensional identity matrix. We assume that the observed data can by described to sufficient accuracy by a linear stationary model

$$0 = A(y - y^{ref}) + Bu$$
$$y^{obs} = y + \epsilon, \tag{1}$$

with $A$ negative definite to ensure stability. Equations of this type typically arise from linear expansion of a non-linear model around a reference state, $y^{ref}$. Linear models are usually preferred for network inference a on larger scale, as the amount of data often limit model complexity and the fact that linear models can give surprisingly good resuits for non-linear cases. The perturbation vector $u$ reflects perturbations that persist long enough to propagate through the network, such as mutations that affect gene activity. Here, $u$ is defined such that for $u = 0$ the system approaches the reference state $y = y^{ref}$. Note that the reference state, $y^{ref}$ is not necessarily the unperturbed state but could be also defined as the average over perturbed and unperturbed states. We assume that the perturbation forces are sampled from a standard normal distribution, with mean $\mathbb{E}[u] = 0$ and covariance matrix $\mathbb{E}[uu^T] = I_N$. The identity matrix is a consequence of the fact that we can absorb the associated standard deviations of the perturbative forces, $u$, in the matrix $B \in \mathbb{R}^{N \times N}$. We introduce normal distributed perturbations for mathematical convenience, as this implies that also $y$ is normal distributed and the resulting maximum likelihood approach is analytically solvable. In general, only the positions of the non-zero elements of $B$ are known from the experimental setup but their actual values are unknown. Using a linear model that operates on log-scale of physical quantities implies that only perturbations can be modelled that act multiplicatively on molecular concentrations. Fortunately, most enzymatic reactions typically fall into this class, such as sequestration and inhibition by other components and also knockout and knockdown experiments can be described on multiplicative level. From Eq. (1) we can derive a relation between the interaction matrix $A$ and the covariance matrix of observed component abundances

$$C := \mathbb{E}\left[\left(y^{obs} - y^{ref}\right)\left(y^{obs} - y^{ref}\right)^T\right]$$
$$= A^{-1}BB^T A^{-T} + \sigma^2 I_N \tag{2}$$

We exploit Eq. (2) to infer directed networks from correlation data. Here, we assume that component abundances are obtained from averaging over a large number of cells. In this case, fast fluctuating perturbations that arise from thermal noise and can be observed only on single-cell level average out. To infer the interaction matrix, $A$, we start with singular value decomposition of the matrix product $A^{-1}B$

$$U\Sigma V^T := A^{-1}B \quad \Rightarrow \quad B = AU\Sigma V^T \tag{3}$$

with $U$ and $V$ orthogonal matrices and $\Sigma$ a diagonal matrix containing the singular values. The negative definite matrix $A$ has full rank and hence is invertible. In the following, we show that it is possible to infer the strength of a directed link between a sender node $j$ and a receiver node $i$, if all direct perturbations on receiver node $i$ are removed from the data set and if a significant partial correlation between $i$ and $j$ exists. Removing the perturbation data for node $i$ implies that the matrix $B$ has at least one zero entry. As a consequence, $N_0 \geq 1$ singular values are zero–as in general not all nodes are perturbed–and the corresponding rows of $U$ span the left null-space of $A^{-1}B$. In the absence of fast fluctuating perturbations, $\gamma = 0$, we can rewrite the covariance matrix as

$$C = A^{-1}BB^T A^{-T} + \sigma^2 I_N \tag{4}$$

$$= U(\Sigma^2 + \sigma^2 I_N)U^T. \tag{5}$$

Assuming that the observed node activities follow a multivariate normal distribution, we can find estimates for the unknown orthogonal matrix $U$, the singular values $\Sigma$, and the observational noise $\sigma$ by maximising the log-likelihood function $\mathcal{L}$ under the constraint $U_l^T U_k = \delta_{lk}$, with $U_k$ the $k$-th column vector of $U$ and $\delta_{lk}$ the Kronecker delta. It fact, it suffices to constrain the norm of the vectors, $\|U_k\|$, as the corresponding maximum likelihood solution leads to an eigenvalue problem with $U_k$ as eigenvectors, which can always be made orthogonal. We can therefore define the likelihood function by

$$\mathcal{L} := \ln \prod_{n=1}^P \mathcal{N}\left(y_n^{obs} | y^{ref}, C\right) + \sum_{k=1}^N \lambda_k \left(U_k^T U_k - 1\right) \tag{6}$$

$$= -\frac{P}{2}\{M \ln 2\pi + \ln|C| + \text{tr}(C^{-1}S)\} + \sum_{k=1}^N \lambda_k\left(U_k^T U_k - 1\right) \tag{7}$$

Here, $S := \frac{1}{P}\sum_{n=1}^{P}\left(y_n^{obs} - y^{ref}\right)\left(y_n^{obs} - y^{ref}\right)^T$ and $y^{ref} := \frac{1}{P}\sum_{n=1}^{P} y_n^{obs}$ denote maximum likelihood estimates of the covariance matrix[37]. From this definition of $y^{ref}$ follows that the initially introduced perturbation vector, $u$, must satisfy, $\frac{1}{P}\sum_{n=1}^{P} u_n = 0$. We further defined with $\lambda_k$ a Lagrange multiplier and denoted by tr(.) the trace of a matrix. In the following calculations, we substitute $S$ by the unbiased sample covariance matrix, $S \to P(P-1)^{-1}S$. Note that $V$ must disappear in the likelihood function as the covariance matrix of $u$ is invariant under any orthogonal transformation $u \to V^T u$.

The maximum of the log-likelihood function is determined by the conditions $d\mathcal{L}/dU_k = 0$, $d\mathcal{L}/d\Sigma_{kk} = 0$, and $d\mathcal{L}/d\sigma^2 = 0$, which results in

$$S\hat{U}_k = \Lambda_k \hat{U}_k \quad \text{with } \Lambda_1 \leq \Lambda_2 \leq \ldots \leq \Lambda_N \qquad (8)$$

$$\hat{\sigma}^2 = \frac{1}{N_0}\sum_{k=1}^{N_0}\Lambda_k \qquad (9)$$

$$\hat{\Sigma}_{kk} = \begin{cases} \sqrt{\Lambda_k - \hat{\sigma}^2} & \text{for } k > N_0 \\ 0 & \text{for } k \leq N_0 \end{cases} \qquad (10)$$

showing that maximum likelihood estimates of $\hat{U}$, $\hat{\sigma}^2$, and $\hat{\Sigma}$ are determined by the sample covariance matrix $S$. If $N_0 > 1$ and the full-rank sample covariance matrix is significantly different from a block-diagonal form—e.g., the network is not separable in subnetworks–the orientations of the corresponding $N_0$ eigenvectors are determined by sampling noise in the space orthogonal to remaining $N - N_0$ eigenvectors. In case that we have less data points than nodes in the network—e.g., the number of perturbed nodes times their replicates is smaller than the network size—some of the $N_0$ smallest eigenvalues become exactly zero and as a consequence the noise level, $\hat{\sigma}$, is underestimated. Although a maximum likelihood solution exists in this case, it is necessary to regularise the covariance matrix, $S \to (1-\epsilon)S + \epsilon I_N$, with $\epsilon$ a regularisation parameter[37], as a correct estimate of the noise level is essential for statistical analysis. Note that the derivation of the maximum likelihood solution is mathematically equivalent to the derivation of principle component analysis from a probabilistic perspective[24].

Solving the matrix equation, Eq. (3), for $A$ gives

$$A = \left(BV\Sigma^+ + W\Sigma^0\right)U^T \qquad (11)$$

with $\Sigma^+$ the pseudoinverse of $\Sigma$. As the matrix $A$ has full rank, we complement $\Sigma^+$ with an unknown diagonal matrix $\Sigma^0$ that has non-zero values where $\Sigma^+$ has zero values and vice versa and complement $BV$ with an unknown orthogonal matrix $W$. Note that by construction, $\Sigma^+ U^T$ and $\Sigma^{+}U^T$ map from complementary subspaces and thereby ensure that $A$ has full rank. The fact that $V$, $W$ and $\Sigma^0$ cannot be determined from $S$ shows that $A$ cannot be computed from a single covariance matrix. A more general case arises when measurement noise is independent but not isotropic, $\sigma^2 I \to \sigma^2 D$, with $D = \text{diag}(r_1, r_2, \ldots, r_N)$ a diagonal matrix with known positive elements that contains scaled noise variances, $r_i = \sigma_i^2/\sigma^2$, resulting in

$$C = A^{-1}BB^T A^{-T} + \sigma^2 D \qquad (12)$$

A transformation to isotropic noise is possible by multiplying both sides of Eq. (12) by $D^{-\frac{1}{2}}$, which changes the result Eq. (11) to

$$A = \left(BV\Sigma^+ + W\Sigma^0\right)U^T D^{-\frac{1}{2}} \qquad (13)$$

with $U$ the eigenvectors of $D^{-\frac{1}{2}}SD^{-\frac{1}{2}}$.

**Case $N_0 = 1$.** We assume that the $i$-th node is the only unperturbed node in the network and hence set $B_{il} = 0$ for all $l$. From Eq. (11) we obtain a unique solution for the $i$-th row of $A$ relative to the diagonal element, $A_{ii}$,

$$\frac{A_{ij}}{A_{ii}} = \frac{\sum_{k,l=1}^{N} B_{il}V_{lk}\Sigma_{kk} + \sum_{k=1}^{N} W_{ik}\Sigma_{kk}^0 U_{kj}^T}{\sum_{k,l=1}^{N} B_{il}V_{lk}\Sigma_{kk} + \sum_{k=1}^{N} W_{ik}\Sigma_{kk}^0 U_{ki}^T} = \left.\frac{U_{kj}^T}{U_{ki}^T}\right|_{k=1} = \frac{U_{j1}}{U_{i1}} \qquad (14)$$

with $U_{j1}$ the $j$-th element of the eigenvector that has the smallest eigenvalue. Note that the first term in the brackets vanishes as $B_{il} = 0$ and $\Sigma_{11}^0$ is the only non-zero element of $\Sigma^0$. The important point is that any dependency on $\sigma$—which affects eigenvalues but not eigenfunctions–has dropped out, making this method asymptotically unbiased with respect to measurement noise. The fact that we can determine the elements of the $i$-th row of $A$ only relative to a reference value, $A_{ii}$, is rooted in fact that we have to determine the $N$ parameters $\{A_{i1}, \ldots, A_{ii}, \ldots, A_{iN}\}$ from $N-1$ perturbations. As a consequence, the strengths of the links onto the target nodes cannot be compared directly if their restoring forces or degradation rates, $A_{ii}$, are different. Generally, only relative values of $A$ can be determined, as the average perturbation strength on node $i$ cannot be disentangled from its restoring force $A_{ii}$–a problem that is typically circumvented by defining $A_{ii} := -1$ for all $i$[7, 13, 15]. For the case that all nodes in the network are perturbed one-by-one, we can cycle through the network and remove the perturbations that act on the current receiver node, whereas keeping the perturbations on the remaining nodes.

By computing the $N$ corresponding covariance matrices and their eigenvectors, we can infer the complete network structure from Eq. (14) if the data quality is sufficiently high. Note that the method makes use of the fact that multi-node perturbations can be realised by superposition of single-node perturbations, which is a special property of linear models.

**Case $N_0 > 1$.** If more than one node are not perturbed we get from Eq. (11)

$$\frac{A_{ij}}{A_{ii}} = \frac{\sum_{k=1}^{N_0} W_{ik}\Sigma_{kk}^0 U_{kj}^T}{\sum_{k=1}^{N_0} W_{ik}\Sigma_{kk}^0 U_{ki}^T} \qquad (15)$$

Non-unique solutions of Eq. (15) can arise if a given fraction of the variance of the receiver node $i$ can be explained by more than one sender node, for example, when a perturbed node $j$ targets two nodes with index $i$ and $l$. In this case it is unclear from the node activity data whether $i$ is affected directly by $j$ or indirectly through $l$, or by a combination of both routes. If node $l$ is not perturbed or only weakly perturbed, a statistical criterion is needed to decide about inferability or identify-ability of the link $j \to i$, which can be computed numerically as follows: To find out whether $j$ transmits a significant amount of information to $i$ that is not passing through $l$, we first remove node $j$ from the observable nodes of the network but keep its perturbative effect on other nodes in the data set. We then determine the link strengths $A'$ for the remaining network of size $N - 1$. To construct a possible realisation of $A'$ we set in Eq. (15) the non-zero values of $\Sigma^0$ to unity and use $W = U$ to arrive at the expression

$$\frac{A'_{il}}{A'_{ii}} = \frac{\sum_{k=1}^{N_0} U'_{ik}U'_{lk}}{\sum_{k=1}^{N_0} U'_{ik}U'_{ik}} \qquad (16)$$

with $U'$ determined from the sample covariance matrix with the $j$-th column and $j$-th row removed. Fixing $W$ and $\Sigma^0$ to seemingly arbitrary values does not affect the result we are after. If $l$ is the only unperturbed node besides $i$, then in the $A'$ system $l$ can now be treated as perturbed—as it may receive perturbations from the unobserved node $j$—and thus Eq. (14) applies. If $l$ is part of many unperturbed nodes that are affected by $j$, then the knowledge how much each of these nodes contributes to the variance of the target node $i$ (which is determined by $W$ and $\Sigma^0$) is irrelevant as we are only interested in the total effect of the alternative routes on node $i$. Using the inferred link strength from Eq. (16) we can rewrite Eq. (2) as a two-node residual inference problem between $j$ and $i$, where we obtain a lower bound for link strength from node $j$ to $i$ by using the variation of $i$ that could not be explained by $A'$. This concept is similar to computing partial correlations. Defining by $\tilde{A}$, $\tilde{B}$ and $\tilde{D}$ the $2 \times 2$ analogues to the full problem we obtain

$$\tilde{C} = \tilde{A}^{-1}BB^T\tilde{A}^{-1} + \sigma^2\tilde{D} \qquad (17)$$

with $\tilde{C}$ the covariance matrix of the vector $\tilde{y}^{obs} = \left(y_j^{obs}, y_i^{obs} + \sum_{l \notin \{i,j\}} A'_{il}(A'_{ii})^{-1}y_l^{obs}\right)^T$ and $\tilde{D}_{11} = r_j$, $\tilde{D}_{22} = r_i + \sum_{l \notin \{i,j\}} A'^2_{il}A'^{-2}_{ii}r_l$, using the scaled variances $r_i = \sigma_i^2/\sigma^2$. Note that $A_{ii} < 0$ for all $i$ as these elements represent sufficiently strong restoring forces that ensure negative definiteness of $A$ and that we have $0 = A'_{ii}y_i^{obs} + \sum_{l \neq i} A'_{il}y_l^{obs}$ from Eq. (1) in the stationary case. An estimate for the minimum relative link strength from node $j$ to node $i$ can be calculated from Eq. (13) and is given by

$$\frac{\tilde{A}_{12}}{\tilde{A}_{11}} = \frac{\tilde{U}_{21}\tilde{D}_{22}^{-1/2}}{\tilde{U}_{11}\tilde{D}_{11}^{-1/2}} \qquad (18)$$

Eq. (18) can be considered as an asymptotically unbiased response coefficient between node 1 as target node and node 2 as source node, as again any dependency on $\sigma^2$ has dropped out. An estimate for the maximum relative link strength from node $j$ to node $i$ follows from Eq. (18) with the off-diagonal elements of $A'$ set to zero. We classify a link as non-inferable if there exists (i) a significant difference between the minimum und maximum estimated link strength and (ii) a minimum link strength that is not significantly different from noise.

**Computational complexity of PRC.** The computational cost for computing PRCs scales as $\mathcal{O}(N_{sub}^3)$, with $N_{sub}$ the size of the subnetwork under consideration. However, as we infer directed networks, we first have to remove the perturbations on each target node before its incoming links can be inferred. The cycling through up to $N_{sub} - 1$ perturbed target nodes increases the computational complexity to $\mathcal{O}(N_{sub}^4)$ in the worst case. As we have generated a subnetwork for perturbed node and used residual bootstrapping to infer statistically significant links, the total computational complexity is given by $\mathcal{O}(N_{boot}N_{per}\langle N_{sub}^4\rangle)$, where $\langle . \rangle$ denotes averaging over all subnetworks, $N_{per}$ the number of perturbed nodes, and $N_{boot}$ the number of bootstrap samples. If the travelling distance of perturbations (correlation length) in the network is significantly shorter than the network diameter, such that $N_{sub}/N \to 0$ in the limit of large networks, $N \to \infty$, the computational complexity scales linearly with network size. In contrast, using Lasso to infer directed links requires $\mathcal{O}(N_{boot}N_{sig}^4)$ operations, as the more efficient Graphical Lasso method[38] is only applicable to undirected networks. Whether our method is computationally

more efficient than Lasso depends on the inference problem. However, for the networks investigated in this work our method required significantly lesscomputational time than inference via Lasso using parallel computing.

**Fraction of inferable links**. Inferability of a directed link between source and target node requires that the remaining network may not contain the same information that is transmitted between them. A sufficient condition is that all information that the remaining network receives from the source node is destroyed by sufficiently strong perturbations. If the target node is not perturbed, information from the source node may reach the remaining network through the target node. In this case also the targets of the target node must be perturbed (Fig. 2a). Counting network motifs that satisfy these conditions gives the number of inferable links. If the network size, $N$, is significantly larger than the number of outgoing links for both the source and target nodes, we can approximate the fraction of inferable links, $F(q)$, by the expression (Supplementary Note 1)

$$F(q) \approx \sum_{k \geq 1} \sum_{l=0} \sum_{m=0}^{\min(k-1,l)} \left[ q^{k+1} + (1-q)q^z \right] P(k,l,m|k \to l)$$

Here, $P(k, l, m|k \to l)$ is the conditional probability of finding two connected nodes in the directed network, where the source node has $k \geq 1$ outgoing links, the target node has $l \geq 0$ outgoing links, and both share $m$ nodes as common targets of their outgoing links. The first term in the brackets corresponds to the case that independent perturbation data for node B exists (Fig. 2a, left panel) and the second term to the case where independent perturbation data for node B are absent (Fig. 2a, right panel). In the calculation of $F(q)$ we assumed that the links in the network are represented by noiseless, linear functions with non-zero slope and that ensure that information of source nodes is neither destroyed nor absorbed in the process of transmission.

**Data preparation**. Kemmeren et al.[3] provided a transcriptome data set of a *Saccharomyces cerevisiae* genome-wide knockout library (with mutant strains isogenic to S288c). This data set comprises transcript levels of 6170 genes for 1484 deletion mutants. The data are presented as the logarithm of the fluorescence intensity ratios (M-values) of transcripts relative to their average abundance across a large number of wild-type replicates, resulting in logarithmic fold changes of mutant/wild-type gene expression levels compared with a wild-type reference level. Kemmeren et al. also used a dye swap setup for several experiments to average out the effect of a possible dye bias. Their chip design measures most of the genes twice per biological sample, thus allowing to estimate the technical variance. The pre-processing of the data is described in Kemmeren et al.[3], Supplementary Information.

**Residual bootstrapping**. We make use of the 748 measured wild-type experimental replicates to determine the natural variation among biological replicates, $\delta_{in} := \log_2(r_{in}) - \langle \log_2(r_{in}) \rangle_n$, with $r_{in} := x_{in}/x_i^{pool}$, $x_{in}$ the expression of gene $i$ in wild-type replicate $n$, $x_i^{pool}$ the expression level of gene $i$ measured after pooling over wild-type replicates, and $\langle . \rangle_n$ denoting the average over replicates. To generate the bootstrap samples we randomly select 200 different $\delta_{in}$ from the replicates for each gene $i$, and add these values to the log fold changes of the perturbed expression levels, $\langle \log_2(r_{in}^{pert}) \rangle_n$, with $r_{in}^{pert} := x_{in}^{pert}/x_i^{pool}$ and the average is taken over the two replicates for each knockout.

**Sparsity constraints by removing noisy nodes**. As network inference typically comes with an insufficient amount of independent perturbations and experimental replicates we run into the problem of overfitting the data. In this case, noisy information from many network nodes is collected to explain the response of a given target node. $L^1$-norm regularised regression (Lasso) systematically removes many links, where each link explains only a small part of the variation of the target node, in favour of few links, where each link contributes significantly. In our approach we remove noisy nodes and thus their potential outgoing links, where the critical noise level is determined by the variability among biological replicates. In the presence of noise, our algorithm removes weakly responding nodes from the network. We thereby assume that the existence of many indirect interactions between source and target node by first distributing the signal of the source node among many weakly responding nodes and then collecting these weak signals to generate a significantly responding target node is much less likely than the existence of a single direct interaction. However, in the noise-free case we run into the same problem as Lasso to determine the right cutoff (regularisation parameter).

**Synthetic data**. Synthetic data were generated using our own model and GeneNetWeaver[25]—an open access software that has been designed for benchmarking network inference methods. With GeneNetWeaver, networks were generated from a model that closely resembles the structure of the yeast regulatory network[25], and steady state levels of node activities were computed using ordinary differential equations. In our data generating model, we first generated scale-free networks with an exponent of 2.5 and an average degree of 2. Then, we solved a system of ordinary differential equations with non-linear regulatory interactions between nodes to obtain steady state values of node activities, e.g., transcript levels. For both models, logarithmic fold changes of node activities were calculated (transcriptional levels upon perturbation relative to wild levels), and gaussian noise was added.

**Code availability**. MATLAB and Python codes for the network inference algorithm and the data preprocessing steps are available on request.

**Data availability**. The data sets analysed during the current study are described in ref.[3] and are available from Gene Expression Omnibus https://www.ncbi.nlm.nih.gov/geo/ under the accession numbers GSE42527, GSE42526, GSE42215, GSE42217, GSE42241 and GSE42240.

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

## Acknowledgements

We acknowledge help from Patrick Kemmeren in interpreting the yeast transcriptome data, the High Performance Computing Plattform of the Heinrich-Heine University and DFG funding by SPP 1395 and the Cluster of Excellence in Plant Sciences (grant no. EXC 1028).

## Author contributions

A.S.K. and M.K. conceived the PRC method. N.H. and M.K. developed the counting procedure for inferable links and the Inferability measure. C.F.B. and M.K. developed the clustering and subnetwork methods and analysed the transcriptome data. N.H. wrote Supplementary Note 1. M.K. wrote Supplementary Note 2. C.F.B. wrote Supplementary Note 3.

## Additional information

**Competing interests:** The authors declare no competing financial interests.

