## [Peer Review File · Nature Communications]

Reviewers' comments:

Reviewer #1 (Remarks to the Author):

Review of "Accounting for biological noise can boost the inferability of transcriptional networks in large-scale gene deletion studies", by C. F. Blum, N. Heramvand, A. S. Khonsari, & M. Kollmann

The authors present a methodology for computing the upper limit of the fraction of inferable links in a network, and a method to infer directed edges in biological networks. The "inferability" estimate is performed using standard combinatorial arguments under strong assumptions. Their inference method is also novel. This is a very thorough and original work, and the authors should be commended for the originality of their work, and the interesting ideas presented in their manuscript. However, I find that some of the assumptions that go in the definition of inferability are not sufficiently justified, that the mathematical formalism of the PRC algorithm is inadequate, and that comparison of their methods with existing benchmarks is lacking. I hope that the authors find my critique of their paper of some utility.

Major Issues

The authors assume that perturbed nodes are randomly distributed within the network. This seems to be a very strong assumption that needs to be justified. For example, targeted therapies in cancer, used as perturbations in many cell line experiments (e.g., Cmap), typically target specific pathways (growth pathways, apoptosis, etc.) How much would the results of the paper change if this assumption didn't hold.

The authors' definition of inferability requires that at least one nodes of each alternative route from A to B be perturbed to make sure that part of the transmitted information is destroyed. However, their computation of $F(q)$ is based on conditions 1 and 2 in Suppl Note 1, which require that the first nodes of the alternative routes [either direct targets of A or direct targets of A (excluding B) and direct targets of B] be perturbed. This seems to be a sufficient but not necessary condition for inferability. In consequence, the estimated fraction of inferred links is not an upper bound (there may be more inferable nodes than counted with conditions 1 and 2). Incidentally, the authors didn't give any expression for $P(k | A B)$, the conditional probability that the source node in a true edge has outdegree k . If I am not mistaken, this conditional probability can be written explicitly in terms of the out-degree distribution as: $P(k | A B) = kP(k)/\langle k \rangle$.

It seems to me that the authors definition of inferability addresses the positive predictive value of an algorithm, by ensuring that if an edge is discovered, it's likely to be a true edge (high precision). However, the definition is not accounting for the recall. A true edge between A and B can be overlooked if the inputs to B from perturbations independent of A overshadow the information coming directly from A to B. This would decrease the sensitivity of the algorithm, and make $F(q)$ not tend to 1 as q tends to 1. To avoid these kind of errors, an additional condition would be necessary: that the incoming edges to B not be perturbed.

Indeed, the authors mention that "networks that are enriched with nodes of high out degree (outgoing hubs) are the most difficult ones to infer (Fig. 1b)." This seems to be at odds with the empirical observation in Marbach et al, Proc Natl Acad Sci U S A. 2010 Apr 6;107(14):6286-91 "Revealing strengths and weaknesses of methods for gene network inference" that nodes with higher in-degree are the most difficult to predict. It would be useful if the authors cast their results in the context of the spectrum of errors for different network motifs discussed in the above mentioned Marbach et al. paper, at least the Fan-in and Fan-out motifs, which seem relevant to their inferability conditions.

I find the discussion leading to the PRC method to be inadequate. First, the authors should explain if the linear expansion of Equation 1 is done in linear scale or in log scale. It seems that it should be in log scale, as the observables are in logarithmic scale. A linearization in log variables or linear variables would result in significantly different interpretations for the coefficients of the matrix A and the nature of the noise (lognormal versus normal). Second, this equation, and its solution (Eq. 2), pertains to the dynamical system that arises in one single cell. In this context, the integration of the noise term in Eq 3 is wrong. It assumed that $\exp\{A(t-t')\} \exp\{A^T(t-t')\} = \exp\{(A+A^T)(t-t')\}$ which is true if A and A^T commute, but not in general. The correct way to find an equation for the covariance matrix in terms of A can be found in Prill et al., Noise-driven causal inference in biomolecular networks, PLoS One. 2015 Jun 1;10(6):e0125777. This error doesn't carry forward in the paper as the authors correctly assume that the η term vanishes after pooling over many cells. Once the cells are pooled, the information of the individual cell dynamics is integrated away, and what remains is the equation for the average over all cells, and how the reference state changes after perturbation. Therefore the computation of the covariance matrix in 3 using the integral of Eq. 2 is misleading and unnecessary: this is not the covariance matrix corresponding to the distribution of states around a fixed point.

The more correct way to deal with the situation that the authors are dealing with seems to simply study how the reference point y_{ref} changes when there is a perturbation. This can be directly done from the averaged equation 1:

$A(\langle y_{\text{new-ref}} \rangle - \langle y_{\text{ref}} \rangle) = -B u$; from where

$$\langle y_{\text{new-ref}} \rangle = \langle y_{\text{ref}} \rangle - A^{-1} B u$$

Then we can add a technical noise term, which is inessential for this discussion. We can always compute the following construct:

$$\langle (\langle y_{\text{new-ref}} \rangle - \langle y_{\text{ref}} \rangle) (\langle y_{\text{new-ref}} \rangle - \langle y_{\text{ref}} \rangle)^T \rangle = A^{-1} B u u^T B^T A^{-T}$$

But we should not confuse this object with the covariance resulting from a dynamical process. Furthermore, the state vector $\langle y_{\text{ref}} \rangle$ of the unperturbed system will not in general be the average over all possible perturbations as assumed in the paper. It is simply the fixed point of the dynamics in the unperturbed system. Furthermore, it is not necessary that the distribution of the perturbations is normally distributed around the unperturbed state $\langle y_{\text{ref}} \rangle$. It is important that the assumption of normality of the perturbed states around the unperturbed state be better justified, as this is an essential component of the formalism (Eq. 7).

I don't understand why the perturbative force u needs to be normally distributed and centered at 0. For a given perturbation, for example overexpression of a gene, the sign of u is always positive. I also don't understand why $E[uu^T] = I_N$. For example, if my perturbation overexpresses nodes 1 and 2, and $u = [u_1 u_2 0 0 \dots]$, the mean values of u_1 and u_2 will not be 0, but a positive number, and the element $\{1,2\}$ of $E[uu^T]$ is $E(u_1)E(u_2)$. If that is not the case, please make a toy example explaining the details of what I may be misunderstanding.

The general perturbations done on a system need not be of the form Bu. They can consist of the modification of the A matrix elements themselves (e.g., a duplication in a promoter regions, the enhancement of degradation by shRNA, etc.). In the general case the equation for the new reference state will be something like:

$$\langle y_{\text{new-ref}} \rangle = (1+A^{-1} \Delta A)^{-1} \langle y_{\text{ref}} \rangle - (1+A^{-1} \Delta A)^{-1} A^{-1} B u$$

where ΔA is the perturbation to the matrix A resulting from a perturbation to the system. The

authors should discuss how general their assumption of that the perturbed state is expressed as BU is.

I find unclear how the choice of W and Σ^0 determine the inference of the elements of the matrix A . For example, in equation 17 W is set to U and the elements of Σ^0 are set to unity. Does this under determination of the method make it prone to gross errors?

It seems intuitive to me that what links are inferable or not depend on the algorithm. For example, from Marbach et al. (Ref 10 in the manuscript) it seems that different algorithms fail in the inference of edges depending on the contextual network motif in which the edge is embedded. Does it make sense then to state that an edge is or is not inferable without stating the nature of the inference algorithm? Along these lines, the excellent agreement between $F(q)$ and the fraction of inferable links resulting from the PRC algorithm shown in Fig 2 of Suppl Note 1 maybe the result that your definition of inferability matches the specific way in which the PRC algorithm works. A discussion on this topic would be appropriate.

Along the same lines, in the DREAM5 benchmark, there were three networks to be inferred. On average across the different methods, the In-silico network was better inferred than the E. coli network which was better inferred than the S. cerevisiae network. It would be useful to check that the inferability I_F of the in-silico network is greater than the I_F of E. coli and greater than the I_F of the yeast network.

Finally, the manuscript is lacking in testing their methods more stringently on real biological data, and in performing comparisons with other network inference methods, especially comparison to different DREAM 3 and 4 Challenges where systematic perturbations were made of individual genes. Such comparison would put the PRC method in the context of other network inference methods. Note that I am not saying that the PRC algorithm must be better than all others. Rather, it would be nice to understand how it can do inferences where other methods can't, and viceversa.

Minor Issues

1. Supplementary Note 1, Eq 1 - the sum should be over c and not m .
2. In the caption to figure 1 shouldn't the second reference of subplot d reference subplot f?
3. There seems to be a typo in Figure 2A, 2nd step diagram titled, "Identify subnetworks". Shouldn't the cluster on the lower left be for nodes 1, 4?
4. There several references to Figure 4, that are seemingly for Figure 2.
5. Suppl 1: line 16: one nodes one node
6. Suppl 1: Line 44: "We therefore dene a inferability measure" "We therefore dene an inferability measure"
7. Suppl 1: line 64 "hubs are souces" "hubs are sources"
8. Adding a clear illustration and small N example of your method will make the paper more readable. Right now I find the paper unclear in terms of how to apply the PRC method.

Reviewer #2 (Remarks to the Author):

The authors of this paper develop analytical and numerical methods to study the problem of inferring transcriptional networks from gene deletion data. The paper has three main components: (1) calculation of the expected fraction of inferable links and study its relation with network structure; (2) development of a method to infer directed networks from perturbation data; (3) imposing network sparsity via a clustering-based procedure. Simulation results have been provided to illustrate their methods. Lastly, they applied their network inference method to publicly available gene knockout data.

Inferring large interaction networks is definitely a problem of current interest. However, I have a few major concerns about the method and results in this work.

I do not follow the authors' reasoning behind the definition of an inferable directed link in the supplementary note 1. The two conditions that a subnetwork must satisfy with respect node perturbation seem too rigid. Is inferability defined by assuming that one can have infinite amount of data? If a link is not inferable, does it mean that parameter associated with the link is not identifiable? Is it related interventional Markov equivalence? It seems that there are a lot more assumptions needed for clear definition of inferability. A rigorous definition should be given.

The partial response coefficients method is developed via maximum likelihood estimate for a Gaussian linear stochastic process. Since the data $y^{\{obs\}}$ is the steady-state observed abundance, what is the advantage of using the dynamic model in (1) than a Gaussian linear structure equation model? The authors did not compare their network inference methods with other competing methods, such as structural equation models. It is hard for the reader appreciate the usefulness of their proposal.

The clustering method for obtaining a sparse network is ad hoc in nature. A more principled way is to impose sparse regularization in their likelihood model and to include delicate variance structure to down-weight noisy nodes. By removing noisy nodes as in the current approach, is it possible that the underlying links will change? Say the true links are $A \rightarrow B \rightarrow C$. If B is removed, then the link becomes $A \rightarrow C$.

In the real data application, it is not clear how to evaluate the authors' method. Some systematic and quantitative evaluations should be provided.

Reviewer #3 (Remarks to the Author):

The paper has two contributions (1) a measure to quantify the "inferability" of regulatory networks and (2) an inference algorithm for such networks.

The two parts of the paper are quite disconnected from each other, or the authors have not made the connection very explicit. One would imagine that the inferability measure would be used to explain the performance of the network inference methods (explain the performance of the new one, but also useful to explain existing ones), but this is not done/explicit.

The writing of the paper is quite disconnected as well. The title, abstract and until mid-page 3 it just talks about inferability. Then out of the blue the network inference method appears, and the inferability is seemingly forgotten.

The flow of the paper is difficult to follow. I don't know if this is because of the style of the journal, but the lack of section headers is not helping at all.

About the actual contributions of the paper, I found both of them interesting and sound. However, their validation/characterisation is quite superficial.

In relation to the inferability measure, it should be studied much more thoroughly. There are many network topology properties but the authors mostly focus only on the out-degree. Why not investigating the influence of e.g. varying centrality in relation to the inferability measure?

In relation to the network inference method, the comparison should be more thorough. Why not using the datasets from the DREAM network inference challenges? (<http://dreamchallenges.org>). They are well known and well-designed experiments, and would allow the comparison of this new

network inference methods against the dozens of methods that participated in the Dream competitions.

<https://www.synapse.org/#!Synapse:syn2787209/wiki/70349>

<https://www.synapse.org/#!Synapse:syn2821735/wiki/71057>

<https://www.synapse.org/#!Synapse:syn1720047/wiki/55342>

In summary, it's promising work, but (1) the two parts of the paper need to be more connected to each other and (2) more thoroughly assessed.

Response to Reviewer's Comments

Response to Reviewer #1:

We very much appreciated the thorough reading by the reviewer, which was extremely helpful for us.

Here our response to the major issues:

- 1) >The authors assume that perturbed nodes are randomly distributed within the network. This seems to be a very strong assumption that needs to be justified ...
 - . In fact, distributing perturbed nodes uniformly over the network has been introduced on the assumption that this may not be a bad approximation for knockout screens. However, as this assumption may not hold in general, we stated now in Supplement 1 that a uniform distribution has been taken for mathematical convenience but in principle any distribution could be used instead. An effect on $F(q)$ is expected if the probability of a node to become a perturbed node correlates with its outdegree. This is now stated in main text (before last sentence in section 'Limits of network inferability').
- 2) > The authors' definition of inferability requires that at least one nodes of each alternative route from A to B be perturbed ... This seems to be a sufficient but not necessary condition for inferability.

In fact, it's a necessary and sufficient condition if we don't make any prior assumption on the network structure. Therefore any node targeted by A could have also an outgoing link to B – which are the shortest indirect routes conceivable. If these nodes are not perturbed, it is not possible to decide whether there exists direct route from A to B or if the information is transmitted by one or more of the shortest indirect routes.

- 3) > It seems to me that the authors definition of inferability addresses the positive predictive value of an algorithm ... However, the definition is not accounting for the recall.

We have to apologise that we have not explicitly stated all conditions under which we calculated $F(q)$. We have included the details now in the main text (first paragraph below the subsection 'Limits of Network Inferability') and in the Supplement 1. The answer to the question is that in computing $F(q)$ we assume that we have an infinite amount of data and that the data does not contain any additional perturbations (such as perturbations arising from thermal noise) and that measurement noise is absent. Consequently, the precision of $F(q)$ is always one (no false positives – that's why we call it an upper bound), as with an (hypothetical) infinite amount of perturbation ex-

periments at hand we can detect an arbitrary small partial correlation between node A and B.

- 4) > the authors mention that “networks that are enriched with nodes of high out degree (outgoing hubs) are the most difficult ones to infer (Fig. 1b) ... This seems to be at odds with the empirical observation that nodes with higher in-degree are the most difficult to predict ...

These two phenomena are disconnected. We have now illustrated the ‘out-degree problem’ in a separate figure (now Fig.~1) as suggested by the reviewers. Here, a high outdegree implies a high number of potential alternative routes, and identifying the true ones comes with the condition that a high number of nodes must be perturbed, even in absence of noise and in presence of an infinite amount of data. This effect is independent of the actual inference method. In contrast, as stated in Marbach et al, PNAS 2010, the fan-in error arises from the fact that the contribution of an incoming link can be mask by the contributions of other incoming links, driving down its signal-to-noise ratio. As consequence, the identification of links pointing to nodes with high indegree strongly depends on the sensitivity on the used algorithm.

- 5) > I find the discussion leading to the PRC method to be inadequate (log scale, A and A^T do not commute in general, definition of the state vector $\langle y_{ref} \rangle$)

We thank the reviewer for having a deep look in the mathematical derivations and spotting the fundamental error in integrating the stochastic differential equation that would be only correct for $A=A^T$. Additionally, we have indeed never mentioned that the equations are defined on log-scale, which is important information. We have corrected these errors in the Methods Section. We also followed the advice of the reviewer and started from stationary equations and directly compute the covariance matrix.

We further followed his advice and clarified the issue in defining $\langle y_{ref} \rangle$, whose interpretation is indeed fixed by the definition of the measured covariance matrix S . As the interpretation of the perturbative noise, u , depends on y_{ref} , we have the situation that the definition of S determines $\langle y_{ref} \rangle$ which in turn determines $\langle u \rangle$.

- 6) > The general perturbations done on a system need not to be of the form Bu . They can consist of the modification of the A matrix elements themselves.

As we work on log scale, the linear model correctly accounts for multiplicative perturbations. More complex perturbations, especially those involving high Hill coefficients (non-constant matrix A) can only be treated approximately. However – as so often – the linear model (on log scale) works surprisingly well in the non-linear case, as shown by our results. We included this statement now in the Methods Section.

- 7) > I find unclear how the choice of W and Σ^0 determine the inference of the elements of the matrix A ... Does this under determination of the method make it prone to gross errors?

In fact, the seemingly arbitrary choice of W and Σ in the Methods Section has no effect on identifying significant links. The reason is that we use

this concept to check if links are non-inferable by trying to explain as much as possible of the variation of the target node (node B) by the remaining nodes in the network with source node (node A) excluded. If we can explain all of the (significant) variation of B we cannot identify if there is a link from A to B. However, the actual link strengths between the nodes in remaining network and node B – which are determined by W and Σ – are of no interest as we want to infer the link from A to B and this is why we can set the matrices to arbitrary but non-zero values. For convenience, we have taken a symmetric solution for W and set Σ to the identity matrix.

8) Does it make sense then to state that an edge is or is not inferable without stating the nature of the inference algorithm?

Inferability or identifiability of link strengths, depends on the model but not on the actual inference algorithm in case of infinite perturbation data. We have to apologize for not clearly stating this fact in the manuscript. The reason is that a link strength is (in principle) identifiable if there exists information that only target node and source node share. If this shared information is sufficient to be detected by an algorithm in presence of noise and limited data is another question and requires statistical analysis.

9) Along these lines, the excellent agreement between $F(q)$ the PRC algorithm maybe the result that your definition of inferability matches the specific way in which the PRC algorithm works.

This is indeed true and we apologize for not stating this in the text. In fact, both methods assume that links are linear functions with non-zero slope. We have stated this in now in the main text and in Supplement 1.

10) Finally, the manuscript is lacking in testing their methods more stringently on real biological data, and in performing comparisons with other network inference methods, especially comparison to different DREAM 3 and 4 Challenges.

We have now tested the performance of our algorithm on the 3 in silico dataset of the DREAM3 Challenge. The surprisingly good performance is now documented in manuscript. We also had a look at the biological data of the Dream5 challenge but for some reason it lacks identities of many perturbation targets, even for knockout mutants, and so our algorithm could not be applied. However, one should have in mind that the 'gold standards' for biological data are likely 'silver standards', as only part of the molecular interactions involved in transcriptional regulation have been experimentally detected so far.

11) Minor Issues

We have correct now these typos in the manuscript.

We are most grateful to the reviewer for spotting all these shortcomings and typos which helped us to improve the manuscript.

Response to Reviewer #2:

- 1) > I do not follow the authors' reasoning behind the definition of an inferable directed link in the supplementary note 1... Is inferability defined by assuming that one can have infinite amount of data? If a link is not inferable, does it mean that parameter associated with the link is not identifiable? Is it related interventional Markov equivalence?

We apologize for not providing a clear definition of inferability before. The definitions now appearing in the main text and in Supplement 1 state that (i) the availability of an infinite amount of data is assumed, (ii) inferability is equivalent to parameter identifiability under the assumption that links are linear function with non-zero slope, and (iii) no prior assumption about the network topology is used (e.g. we don't assume that the network is a DAG). As we allow for feedbacks we don't see a direct relation to interventional Markov equivalence, which is based on structural equation models under the assumption that the network belongs to the class of DAGs.

- 2) > What is the advantage of using the dynamic model in (1) than a Gaussian linear structure equation model?

We agree with the reviewer that a dynamical approach is unnecessary here and we now use a linear Gaussian model (see Methods Section).

- 3) >The authors did not compare their network inference methods with other competing methods, such as structural equation models. It is hard for the reader appreciate the usefulness of their proposal.

We now tested our approach on the DREAM3 challenge data, which suits our approach as this dataset provides the complete target identities of the knockout mutants (which is not the case for the DREAM5 challenge) Although we did not use dynamical data for inference and used the same fixed (but Bonferroni corrected) significance level for all data set, we achieved 2nd place among the 29 submitted network inference methods. We report this result now in the main text.

- 4) > in the real data application, it is not clear how to evaluate the authors' method. Some systematic and quantitative evaluations should be provided

For real data applications, such analyzing transcriptome data for *E. coli* and yeast one should have in mind that the 'gold standards' are more 'silver standards' in reality, as only part of the molecular interactions involved in transcriptional regulation have been measured. As such we find it difficult to assess the performance of an algorithm on real data, as performance is to our experience strongly affected by the data preprocessing steps that are used prior to analysis.

Response to Reviewer #3:

- 1) The two parts of the paper are quite disconnected from each other .. and the writing of the paper is quite disconnected as well.

We now tried to connect both parts in better way. However, as the $F(q)$ appears in the plots of the inference part for comparison and makes the power of structural constraints explicit, there is a clear connection, at least content-related.

- 2) There are many network topology properties but the authors mostly focus only on the out-degree. Why not investigating the influence of e.g varying centrality in relation to the inferability measure?

The reason is that our inferability measure, $F(q)$, depends essentially only on the outdegree of nodes. As such, comparing $F(q)$ or its integral with other network measures would just reflect their relation to the outdegree distribution. It is also difficult to find a meaningful comparison between $F(q)$, which is associated with the local network structure around given links, to centrality, which is a node specific property associated with the global network structure.

- 3) In relation to the network inference method, the comparison should be more thorough. Why not using the datasets from the DREAM network inference challenges?

We apologize for not having done this comparison earlier. We now tested our approach on the DREAM3 challenge data, which suits our approach as this dataset provides the complete target identities of the knockout mutants (which is not the case for the DREAM5 challenge)

Although we did not use dynamical data for inference and used the same fixed (but Bonferroni corrected) significance level for all data sets, we achieved 2nd place among the 29 submitted network inference methods. We reported this result now in the main text.

Reviewers' comments:

Reviewer #1 (Remarks to the Author):

The authors have answered our concerns to a great extent, but some concerns remain. On the positive side, we have found their contributions to the network inference community, namely, the counting of inferable edges and their inference strategy to be of interest and original. However, we have found the authors' presentation and construction of central arguments to be somewhat loose.

Two examples are as follows. First, the authors' present an interesting strategy for computing the average fraction of inferable edges, provided that strict conditions are met. However, the connection of this calculation to the inference strategy is not sufficiently emphasized such that the work appears integrated. Second, in our opinion designating variability in biological replicates as biological "noise" is misleading. The variability of measurements are likely not attributable to any fluctuations that are intrinsic to the system being studied, but are manifestations of the imprecision of the experimental conditions. We would reserve the phrase "biological noise" for true stochasticity in the system under study (e.g., fluctuations in transcription of a gene due to the stochastic nature of the dynamics). Taken together, we are concerned that the novelty of the authors' work may not be fully appreciated by the reader due to the presentation.

Reviewer #2 (Remarks to the Author):

The authors have made major revisions, such as replacing their dynamic model by a more transparent Gaussian linear model for estimating network interaction matrices. The revised paper is in general more readable. Here are some remaining comments:

First, some technical comments on the MLE from Eq (6) to (10).

a) The authors state that U should be an orthogonal matrix. But the constraints in (6) are only imposed on the norm of U_k . Should one also include constraints like $U_k^t U_j = 0$ so that the off-diagonal elements of $U^t U$ are zero?

b) Does the MLE still exist if the sample covariance matrix S is not positive definite? This happens if there are not sufficient data points compared to the number of nodes in a large network. In particular, (8) will not lead to unique U_k , since the eigenvectors of S are not unique.

c) To improve the readability of this part, I suggest the authors differentiate estimates from parameters by using $\hat{\cdot}$ on estimates. The current notation is a little confusing.

Second, on network inferability. According to my reading, the network inferability part is mostly an empirical study on the relation between some features of network topology and the fraction of inferable/identifiable edge links. This gives some general idea about average inferability, but does not answer the question about the identifiability for a particular class of underlying networks. The authors should clarify this difference by referring to those works on causal network identifiability from intervention data, e.g. Fu and Zhou (JASA 2013, 108: 288-300).

Lastly, a minor point. The probability $P(k,l,m|k \rightarrow l)$ appears on page 5, but its definition is given in Methods which comes much later.

Reviewer #3 (Remarks to the Author):

The authors have addressed my concerns.

However, it was a consistent request from all three reviewers that the authors use data/networks from the DREAM challenges to evaluate their methods. This has been done partially, only using data from the DREAM3 competition, because the data from other challenges is not suitable for this method due to the lack of known perturbation targets.

This makes me wonder about the scope of applicability of this method. What is its real value if it can only be applied to a very specific set of datasets? The authors should explicitly address the question of the scope of applicability of their method in the discussion section.

Response to Reviewer #1:

- First, the authors' present an interesting strategy for computing the average fraction of inferable edges, provided that strict conditions are met. However, the connection of this calculation to the inference strategy is not sufficiently emphasized such that the work appears integrated ...

We have to admit that we have never mentioned in the ms that the central idea of our network inference method (to a priori reduce the number of alternative routes from source to target e.g. subnetwork method) is a direct consequence of the inferability analysis done in the first half of the paper. We therefore addressed this issue:

Line 34: *"In this work we first investigate the causes that affect network inferability by introducing a simple network inferability measure and subsequently use the gained insight to design an unbiased, scalable network inference algorithm."*

Line 108: *"From our analyses of network inferability we gained the insight that the number of potential alternative routes how a source node can affect a target node correlates positively with the outdegree of the source node and inversely with the expected inferability of the directed link between source and target, given that perturbed nodes are uniformly distributed in the network. Consequently, network inference algorithms should strongly benefit from an a priori reduction in the number of alternative routes. In the following we present an unbiased network inference algorithm that eliminates alternative routes with low signal-to-noise ratio as a preprocessing step."*

- Second, in our opinion designating variability in biological replicates as biological "noise" is misleading. The variability of measurements are likely not attributable to any fluctuations that are intrinsic to the system being studied, but are manifestations of the imprecision of the experimental conditions. We would reserve the phrase "biological noise" for true stochasticity in the system ...

We agree with the reviewer on this issue. We therefore changed the misleading expression 'biological noise' to 'variability among biological replicates'.

Response to Reviewer #2:

- a) The authors state that U should be an orthogonal matrix. But the constraints in (6) are only imposed on the norm of U_k . Should one also include constraints like $U_k^t U_j = 0$ so that the off-diagonal elements of $U^t U$ are zero?

There is indeed some imprecision in our formulation. We now give an explanation (Line 399 onwards) that this constraint is sufficient as the vectors of the resulting eigenvalue problem can always be made orthogonal. Although we are aware that this step can be formulated in a more rigorous way, we would like to keep this 'shortcut' as it keeps the notation simple.

- b) Does the MLE still exist if the sample covariance matrix S is not positive definite? This happens if there are not sufficient data points compared to the number of nodes in a large network. In particular, (8) will not lead to unique U_k , since the eigenvectors of S are not unique.

This is a clever question, which we have not addressed yet. The short answer is that MLE solution exists but it might make not much sense. The long answer is given from line 407 onwards.

- c) To improve the readability of this part, I suggest the authors differentiate estimates from parameters by using $\hat{\cdot}$ on estimates. The current notation is a little confusing.

We now included this notation

- ... According to my reading, the network inferability part is mostly an empirical study on the relation between some features of network topology and the fraction of inferable/identifiable edge links. This gives some general idea about average inferability, but does not answer the question about the identifiability for a particular class of underlying networks. The authors should clarify this difference by referring to those works on causal network identifiability from intervention data, e.g. Fu and Zhou (JASA 2013, 108: 288-300).

We now address this issue and included the reference from line 74 onwards.

- Lastly, a minor point. The probability $P(k,l,m|k \rightarrow l)$ appears on page 5, but its definition is given in Methods which comes much later.

We apologize for this error. The probability comes now with definition on line 100.

Response to Reviewer #2:

- ... because the data from other challenges is not suitable for this method due to the lack of known perturbation targets. This makes me wonder about the scope of applicability of this method. What is its real value if it can only be applied to a very specific set of datasets?

In fact, some of the more recent DREAM challenges for network inference include a large amount of knockout data but for some reason the organizers don't give the information which gene was knocked out to the participants. One should also note that the knockout and knockdown libraries with high coverage have only recently been established in other model organisms than yeast, thanks to novel gene editing methods. As targeted gene editing methods become more widespread, we believe it is just a matter of time until the transcriptome data will follow.

We have discussed this issue now from line 266 onwards.

REVIEWERS' COMMENTS:

Reviewer #2 (Remarks to the Author):

The authors have addressed my comments. I recommend acceptance.